# RETENTIVE NETWORK: A SUCCESSOR TO TRANSFORMER FOR LARGE LANGUAGE MODELS

## ABSTRACT

In this work, we propose **Retentive Network** (RETNET) as a foundation architecture for large language models, simultaneously achieving training parallelism, low-cost inference, and good performance. We theoretically derive the connection between recurrence and attention. Then we propose the retention mechanism for sequence modeling, which supports three computation paradigms, i.e., parallel, recurrent, and chunkwise recurrent. Specifically, the parallel representation allows for training parallelism. The recurrent representation enables low-cost $O(1)$ inference, which improves decoding throughput, latency, and GPU memory without sacrificing performance. The chunkwise recurrent representation facilitates efficient long-sequence modeling with linear complexity, where each chunk is encoded parallelly while recurrently summarizing the chunks. Experimental results on language modeling show that RETNET achieves favorable scaling results, parallel training, low-cost deployment, and efficient inference. The intriguing properties make RETNET a strong successor to Transformer for large language models.

## 1 INTRODUCTION

Transformer (Vaswani et al., 2017) has become the de facto architecture for large language models, which was initially proposed to overcome the sequential training issue of recurrent models (Hochreiter & Schmidhuber, 1997). However, training parallelism of Transformers is at the cost of inefficient inference, because of the $O(N)$ complexity per step and memory-bound key-value cache (Shazeer, 2019), which renders Transformers unfriendly to deployment. The growing sequence length increases GPU memory consumption as well as latency and reduces inference speed. Numerous efforts have continued to develop the next-generation architecture, aiming at retaining training parallelism and competitive performance as Transformers while having efficient $O(1)$ inference. It is challenging to achieve the above goals simultaneously, i.e., the so-called "impossible triangle" as shown in Figure 1.

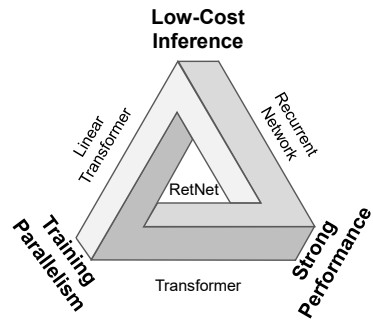

Figure 1: RetNet makes the "impossible triangle" possible, which achieves training parallelism, good performance, and low inference cost simultaneously.

There have been three main strands of research. First, linearized attention (Katharopoulos et al., 2020) approximates standard attention scores $\exp(\boldsymbol{q} \cdot \boldsymbol{k})$ with kernels $\phi(\boldsymbol{q}) \cdot \phi(\boldsymbol{k})$, so that autoregressive inference can be rewritten in a recurrent form. However, the modeling capability and performance are worse than Transformers, which hinders the method's popularity. The second strand returns to recurrent models for efficient inference while sacrificing training parallelism. As a remedy, element-wise operators (Peng et al., 2023) are used for acceleration, however, representation capacity and performance are harmed. The third line explores replacing attention with other mechanisms, such as S4 (Gu et al., 2021), and its variants (Dao et al., 2022b; Poli et al., 2023). None of the previous work can break through the impossible triangle, resulting in no clear winner compared with Transformers.

In this work, we propose retentive networks (RetNet), achieving low-cost inference, efficient long-sequence modeling, Transformer-comparable performance, and parallel model training simultaneously. Specifically, we introduce a multi-scale retention mechanism to substitute multi-head attention,

which has three computation paradigms, i.e., parallel, recurrent, and chunkwise recurrent representations. First, the parallel representation empowers training parallelism to utilize GPU devices fully. Second, the recurrent representation enables efficient $O(1)$ inference in terms of memory and computation. The deployment cost and latency can be significantly reduced. Moreover, the implementation is greatly simplified without key-value cache tricks. Third, the chunkwise recurrent representation can perform efficient long-sequence modeling. We parallelly encode each local block for computation speed while recurrently encoding the global blocks to save GPU memory.

We compare RetNet with Transformer and its variants. Experimental results on language modeling show that RetNet is consistently competitive in terms of both scaling curves and in-context learning. Moreover, the inference cost of RetNet is length-invariant. For a 7B model and 8k sequence length, RetNet decodes $8.4\times$ faster and saves 70% of memory than Transformers with key-value caches. During training, RetNet also achieves 25-50% memory saving and $7\times$ acceleration than standard Transformer and an advantage towards highly-optimized FlashAttention (Dao et al., 2022a). Besides, RetNet's inference latency is insensitive to batch size, allowing enormous throughput. The intriguing properties make RetNet a strong successor to Transformer for large language models.

## 2 RETENTIVE NETWORKS

Retentive network (RetNet) is stacked with $L$ identical blocks, which follows a similar layout (i.e., residual connection, and pre-LayerNorm) as in Transformer (Vaswani et al., 2017). Each RetNet block contains two modules: a multi-scale retention (MSR) module, and a feed-forward network (FFN) module. We introduce the MSR module in the following sections. Given an input sequence $x = x_1 \cdots x_{|x|}$, RetNet encodes the sequence in an autoregressive way. The input vectors $\{\boldsymbol{x}_i\}_{i=1}^{|x|}$ is first packed into $X^0 = [\boldsymbol{x}_1, \cdots, \boldsymbol{x}_{|x|}] \in \mathbb{R}^{|x| \times d_{\text{model}}}$, where $d_{\text{model}}$ is hidden dimension. Then we compute contextualized vector representations $X^l = \text{RetNet}_l(X^{l-1}), l \in [1, L]$.

### 2.1 RETENTION

In this section, we introduce the retention mechanism that has a dual form of recurrence and parallelism. So we can train the models in a parallel way while recurrently conducting inference.

Given input $X \in \mathbb{R}^{|x| \times d_{\text{model}}}$, we project it to one-dimensional function $v(n) = X_n \cdot \boldsymbol{w}_V$. Consider a sequence modeling problem that maps $v(n) \mapsto o(n)$ through states $\boldsymbol{s}_n$. Let $v_n, o_n$ denote $v(n), o(n)$ for simplicity. We formulate the mapping in a recurrent manner:

$$
\begin{aligned}
\boldsymbol{s}_n &= A\boldsymbol{s}_{n-1} + K_n^{\mathsf{T}} v_n, & A \in \mathbb{R}^{d \times d}, K_n \in \mathbb{R}^{1 \times d} \\
o_n &= Q_n \boldsymbol{s}_n = \sum_{m=1}^{n} Q_n A^{n-m} K_m^{\mathsf{T}} v_m, & Q_n \in \mathbb{R}^{1 \times d}
\end{aligned}
\tag{1}
$$

where we map $v_n$ to the state vector $\boldsymbol{s}_n$, and then implement a linear transform to encode sequence information recurrently. Next, we make the projection $Q_n, K_n$ content-aware:

$$
Q = XW_Q, \quad K = XW_K
\tag{2}
$$

where $W_Q, W_K \in \mathbb{R}^{d \times d}$ are learnable matrices.

We diagonalize the matrix $A = \Lambda(\gamma e^{i\theta})\Lambda^{-1}$, where $\gamma, \theta \in \mathbb{R}^d$. Then we obtain $A^{n-m} = \Lambda(\gamma e^{i\theta})^{n-m}\Lambda^{-1}$. By absorbing $\Lambda$ into $W_Q$ and $W_K$, we can rewrite Equation (1) as:

$$
\begin{aligned}
o_n &= \sum_{m=1}^{n} Q_n (\gamma e^{i\theta})^{n-m} K_m^{\mathsf{T}} v_m \\
&= \sum_{m=1}^{n} (Q_n (\gamma e^{i\theta})^n)(K_m (\gamma e^{i\theta})^{-m})^{\mathsf{T}} v_m
\end{aligned}
\tag{3}
$$

where $Q_n(\gamma e^{i\theta})^n, K_m(\gamma e^{i\theta})^{-m}$ is known as xPos (Sun et al., 2023), i.e., a relative position embedding proposed for Transformer. We further simplify $\gamma$ as a scalar, Equation (3) becomes:

$$
o_n = \sum_{m=1}^{n} \gamma^{n-m} (Q_n e^{in\theta})(K_m e^{im\theta})^{\dagger} v_m
\tag{4}
$$

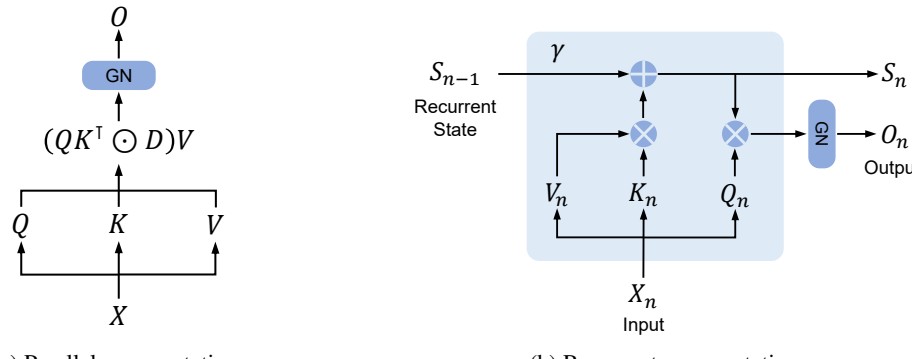

Figure 2: Dual form of RetNet. "GN" is short for GroupNorm.

where $^\dagger$ is the conjugate transpose. The formulation is easily parallelizable within training instances.

In summary, we start with recurrent modeling as shown in Equation (1), and then derive its parallel formulation in Equation (4). We consider the original mapping $v(n) \mapsto o(n)$ as vectors and obtain the retention mechanism as follows.

**The Parallel Representation of Retention**    As shown in Figure 2a, the retention layer is defined as:

$$Q = (XW_Q) \odot \Theta, \quad K = (XW_K) \odot \overline{\Theta}, \quad V = XW_V$$

$$\Theta_n = e^{in\theta}, \quad D_{nm} = \begin{cases} \gamma^{n-m}, & n \geq m \\ 0, & n < m \end{cases} \tag{5}$$

$$\text{Retention}(X) = (QK^\mathsf{T} \odot D)V$$

where $D \in \mathbb{R}^{|x| \times |x|}$ combines causal masking and exponential decay along relative distance as one matrix, and $\overline{\Theta}$ is the complex conjugate of $\Theta$. In practice, we map $Q, K \in \mathbb{R}^d \to \mathbb{C}^{d/2}$, add the complex position embedding $\Theta$, then map them back to $\mathbb{R}^d$, following the implementation trick as in LLaMA (Touvron et al., 2023a; Su et al., 2021). Similar to self-attention, the parallel representation enables us to train the models with GPUs efficiently.

**The Recurrent Representation of Retention**    As shown in Figure 2b, the proposed mechanism can also be written as recurrent neural networks (RNNs), which is favorable for inference. For the $n$-th timestep, we recurrently obtain the output as:

$$S_n = \gamma S_{n-1} + K_n^\mathsf{T} V_n$$
$$\text{Retention}(X_n) = Q_n S_n, \quad n = 1, \cdots, |x| \tag{6}$$

where $Q, K, V, \gamma$ are the same as in Equation (5).

**The Chunkwise Recurrent Representation of Retention**    A hybrid form of parallel representation and recurrent representation is available to accelerate training, especially for long sequences. We divide the input sequences into chunks. Within each chunk, we follow the parallel representation (Equation (5)) to conduct computation. In contrast, cross-chunk information is passed following the recurrent representation (Equation (6)). Specifically, let $B$ denote the chunk length. We compute the retention output of the $i$-th chunk via:

$$Q_{[i]} = Q_{Bi:B(i+1)}, \quad K_{[i]} = K_{Bi:B(i+1)}, \quad V_{[i]} = V_{Bi:B(i+1)}$$

$$R_i = K_{[i]}^\mathsf{T}(V_{[i]} \odot \zeta) + \gamma^B R_{i-1}, \quad \zeta_{ij} = \gamma^{B-i-1}$$

$$\text{Retention}(X_{[i]}) = \underbrace{(Q_{[i]} K_{[i]}^\mathsf{T} \odot D)V_{[i]}}_{\text{Inner-Chunk}} + \underbrace{(Q_{[i]} R_{i-1}) \odot \xi}_{\text{Cross-Chunk}}, \quad \xi_{ij} = \gamma^{i+1} \tag{7}$$

where $[i]$ indicates the $i$-th chunk, i.e., $x_{[i]} = [x_{(i-1)B+1}, \cdots, x_{iB}]$.

## 2.2 GATED MULTI-SCALE RETENTION

We use $h = d_{\text{model}}/d$ retention heads in each layer, where $d$ is the head dimension. The heads use different parameter matrices $W_Q, W_K, W_V \in \mathbb{R}^{d \times d}$. Moreover, **m**ulti-**s**cale **r**etention (MSR) assigns different $\gamma$ for each head. For simplicity, we set $\gamma$ identical among different layers and keep them fixed. In addition, we add a swish gate (Hendrycks & Gimpel, 2016; Ramachandran et al., 2017) to increase the non-linearity of retention layers. Formally, given input $X$, we define the layer as:

$$
\begin{aligned}
\gamma &= 1 - 2^{-5-\text{arange}(0,h)} \in \mathbb{R}^h \\
\text{head}_i &= \text{Retention}(X, \gamma_i) \\
Y &= \text{GroupNorm}_h(\text{Concat}(\text{head}_1, \cdots, \text{head}_h)) \\
\text{MSR}(X) &= (\text{swish}(XW_G) \odot Y)W_O
\end{aligned}
\tag{8}
$$

where $W_G, W_O \in \mathbb{R}^{d_{\text{model}} \times d_{\text{model}}}$ are learnable parameters, and GroupNorm (Wu & He, 2018) normalizes the output of each head, following SubLN proposed in (Shoeybi et al., 2019). Notice that the heads use multiple $\gamma$ scales, which results in different variance statistics. So we normalize the head outputs separately. The pseudocode of retention is summarized in Appendix D.

**Retention Score Normalization** We utilize the scale-invariant nature of GroupNorm to improve numerical precision of retention layers. Specifically, multiplying a scalar value within GroupNorm does not affect outputs and backward gradients, i.e., $\text{GroupNorm}(\alpha * \text{head}_i) = \text{GroupNorm}(\text{head}_i)$. We implement three normalization factors in Equation (5). First, we normalize $QK^\intercal$ as $QK^\intercal/\sqrt{d}$. Second, we replace $D$ with $\tilde{D}_{nm} = D_{nm}/\sqrt{\sum_{i=1}^n D_{ni}}$. Third, let $R$ denote the retention scores $R = QK^\intercal \odot D$, we normalize it as $\tilde{R}_{nm} = R_{nm}/\max(|\sum_{i=1}^n R_{ni}|, 1)$. Then the retention output becomes $\text{Retention}(X) = \tilde{R}V$. The above tricks do not affect the final results while stabilizing the numerical flow of both forward and backward passes, because of the scale-invariant property.

## 2.3 OVERALL ARCHITECTURE OF RETENTION NETWORKS

For an $L$-layer retention network, we stack multi-scale retention (MSR) and feed-forward network (FFN) to build the model. Formally, the input sequence $\{x_i\}_{i=1}^{|x|}$ is transformed to vectors by a word embedding layer. We use the packed embeddings $X^0 = [\boldsymbol{x}_1, \cdots, \boldsymbol{x}_{|x|}] \in \mathbb{R}^{|x| \times d_{\text{model}}}$ as the input and compute the model output $X^L$:

$$
\begin{aligned}
Y^l &= \text{MSR}(\text{LN}(X^l)) + X^l \\
X^{l+1} &= \text{FFN}(\text{LN}(Y^l)) + Y^l
\end{aligned}
\tag{9}
$$

where $\text{LN}(\cdot)$ is LayerNorm (Ba et al., 2016). The FFN part is computed as $\text{FFN}(X) = \text{gelu}(XW_1)W_2$, where $W_1, W_2$ are parameter matrices.

**Training** We use the parallel (Equation (5)) and chunkwise recurrent (Equation (7)) representations during the training process. The parallelization within sequences or chunks efficiently utilizes GPUs to accelerate computation. More favorably, chunkwise recurrence is especially useful for long-sequence training, which is efficient in terms of both FLOPs and memory consumption.

**Inference** The recurrent representation (Equation (6)) is employed during the inference, which nicely fits autoregressive decoding. The $O(1)$ complexity reduces memory and inference latency while achieving equivalent results.

## 2.4 RELATION TO AND DIFFERENCES FROM PREVIOUS METHODS

Table 1 compares RetNet with previous methods from various perspectives. The comparison results echo the "impossible triangle" presented in Figure 1. Moreover, RetNet has linear memory complexity for long sequences due to the chunkwise recurrent representation. We also summarize the comparisons with specific methods as follows.

| Architectures | Training Parallelization | Inference Cost | Long-Sequence Memory Complexity | Performance |
|---|---|---|---|---|
| Transformer | ✔ | $O(N)$ | $O(N^2)$ | ✔✔ |
| Linear Transformer | ✔ | $O(1)$ | $O(N)$ | ✘ |
| Recurrent NN | ✘ | $O(1)$ | $O(N)$ | ✘ |
| RWKV | ✘* | $O(1)$ | $O(N)$ | ✔ |
| H3/S4 | ✔ | $O(1)$ | $O(N \log N)$ | ✔ |
| Hyena | ✔ | $O(N)$ | $O(N \log N)$ | ✔ |
| RetNet | ✔ | $O(1)$ | $O(N)$ | ✔✔ |

Table 1: Model comparison from various perspectives. The inference cost is measured as one-step inference complexity. RetNet achieves training parallelization, constant inference cost, linear long-sequence memory complexity, and good performance. "∗": whether the training implementation is sequentially parallelized, although RWKV uses channel-wise parallelism.

**Transformer** The parallel representation of retention shares similar spirits as Transformers (Vaswani et al., 2017). The most related Transformer variant is Lex Transformer (Sun et al., 2023) which implements xPos as position embeddings. As described in Equation (3), the derivation of retention aligns with xPos. In comparison with attention, retention removes $\mathrm{softmax}$ and enables recurrent formulation, which significantly benefits inference.

**S4** Unlike Equation (2), if $Q_n$ and $K_n$ are content-unaware, the formulation can be degenerated to S4 (Gu et al., 2021), where $O = (QK^\intercal, QAK^\intercal, .., QA^{|x|-1}K^\intercal) * V$.

**Linear Attention** The variants typically use various kernels $\phi(q_i)\phi(k_j)/\sum_{n=1}^{|x|} \phi(q_i)\phi(k_n)$ to replace the $\mathrm{softmax}$ function. However, linear attention struggles to effectively encode position information, rendering the models less performant. Besides, we reexamine sequence modeling from scratch, rather than aiming at approximating $\mathrm{softmax}$.

**AFT/RWKV** Attention Free Transformer (AFT) simplifies dot-product attention to element-wise operations and moves $\mathrm{softmax}$ to key vectors. RWKV replaces AFT's position embeddings with exponential decay and runs the models recurrently for training and inference. In comparison, retention preserves high-dimensional states to encode sequence information, which contributes to expressive ability and better performance.

**xPos/RoPE** Compared with relative position embedding methods proposed for Transformers, Equation (3) presents a similar formulation as xPos (Sun et al., 2023) and RoPE (Su et al., 2021).

**Sub-LayerNorm** As shown in Equation (8), the retention layer uses Sub-LayerNorm (Wang et al., 2022b) to normalize outputs. Because the multi-scale modeling leads to different variances for the heads, we replace the original LayerNorm with GroupNorm.

## 3 EXPERIMENTS

We conduct experiments on language modeling to evaluate RetNet. We evaluate the proposed architecture with language modeling performance and zero-/few-shot learning on downstream tasks. Moreover, for training and inference, we compare speed, memory consumption, and latency.

### 3.1 SETUP

**Parameter Allocation** We re-allocate the parameters in MSR and FFN for fair comparisons. Let $d$ denote $d_{\text{model}}$ for simplicity here. In Transformers, there are about $4d^2$ parameters in self-attention where $W_Q, W_K, W_V, W_O \in \mathbb{R}^{d \times d}$, and $8d^2$ parameters in FFN where the intermediate dimension is $4d$. In comparison, RetNet has $8d^2$ parameters in retention, where $W_Q, W_K \in \mathbb{R}^{d \times d}, W_G, W_V \in \mathbb{R}^{d \times 2d}, W_O \in \mathbb{R}^{2d \times d}$. Notice that the head dimension of $V$ is twice $Q, K$. The widened dimension is

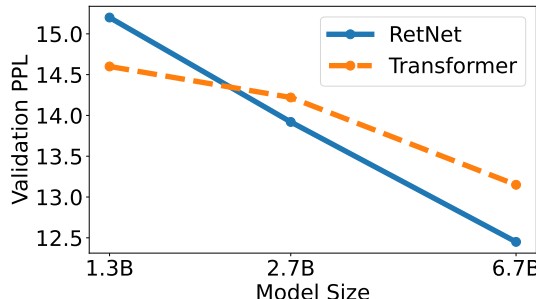

Figure 3: Perplexity decreases along with scaling up the model size. We empirically observe that RetNet tends to outperform Transformer when the model size is larger than 2B.

| | HS | BoolQ | COPA | PIQA | Winograd | Winogrande | SC | Avg |
|---|---|---|---|---|---|---|---|---|
| *Zero-Shot* | | | | | | | | |
| Transformer | 55.9 | 62.0 | 69.0 | 74.6 | 69.5 | 56.5 | 75.0 | 66.07 |
| RetNet | **60.7** | **62.2** | **77.0** | **75.4** | **77.2** | **58.1** | **76.0** | **69.51** |
| *4-Shot* | | | | | | | | |
| Transformer | 55.8 | 58.7 | 71.0 | 75.0 | 71.9 | 57.3 | 75.4 | 66.44 |
| RetNet | **60.5** | **60.1** | **78.0** | **76.0** | **77.9** | **59.9** | **75.9** | **69.76** |

Table 2: Zero-shot and few-shot learning with Transformer and RetNet. The model size is 6.7B.

projected back to $d$ by $W_O$. In order to keep the parameter number the same as Transformer, the FFN intermediate dimension in RetNet is $2d$. Meanwhile, we set the head dimension to 256, i.e., 256 for queries and keys, and 512 for values. For fair comparison, we keep $\gamma$ identical among different model sizes, where $\gamma = 1 - e^{\text{linspace}(\log 1/32, \log 1/512, h)} \in \mathbb{R}^h$ instead of the default value in Equation (8).

**Language Model Training**    We train language models with various sizes (i.e., 1.3B, 2.7B, and 6.7B) from scratch. The hyper-parameters are attached in Appendix A. The training corpus is a curated compilation of The Pile (Gao et al., 2020), C4 (Dodge et al., 2021), and The Stack (Kocetkov et al., 2022). We append the `<bos>` token to indicate the start of a sequence[1]. The training batch size is 4M tokens with 2048 maximal length. We train the models with 100B tokens, i.e., 25k steps. We use the AdamW (Loshchilov & Hutter, 2019) optimizer with $\beta_1 = 0.9, \beta_2 = 0.98$, and weight decay is set to 0.05. The number of warmup steps is 375 with linear learning rate decay. The parameters are initialized following DeepNet (Wang et al., 2022a) to guarantee training stability. The implementation is based on TorchScale (Ma et al., 2022). We train the models with 512 AMD MI200 GPUs.

### 3.2    COMPARISONS WITH TRANSFORMER

**Language Modeling**    As shown in Figure 3, we report perplexity on the validation set for the language models based on Transformer and RetNet. We present the scaling curves with three model sizes, i.e., 1.3B, 2.7B, and 6.7B. RetNet achieves comparable results with Transformers. More importantly, the results indicate that RetNet is favorable regarding size scaling. Besides performance, the RetNet training is quite stable in our experiments. Experimental results show that RetNet is a strong competitor to Transformer for large language models. Empirically, we find that RetNet starts to outperform Transformer when the model size is larger than 2B. We also summarize the language modeling results with different context lengths in Appendix B.

**Zero-Shot and Few-Shot Evaluation on Downstream Tasks**    We also compare the language models on a wide range of downstream tasks. We evaluate zero-shot and 4-shot learning with the 6.7B models. As shown in Table 2, the datasets include HellaSwag (HS; Zellers et al. 2019), BoolQ (Clark et al., 2019), COPA (Wang et al., 2019), PIQA (Bisk et al., 2020), Winograd, Winogrande (Levesque

---

[1]We find that appending the `<bos>` token at the beginning benefits training stability and performance.

| Model Size | Memory (GB) ↓ | | | Throughput (wps) ↑ | | |
|---|---|---|---|---|---|---|
| | Trm | Trm+FlashAttn | RetNet | Trm | Trm+FlashAttn | RetNet |
| 1.3B | 74.8 | 38.8 | 34.5 | 10832.4 | 63965.2 | 73344.8 |
| 2.7B | 69.6 | 42.1 | 42.0 | 5186.0 | 34990.2 | 38921.2 |
| 6.7B | 69.0 | 51.4 | 48.0 | 2754.4 | 16230.1 | 17458.6 |
| 13B | 61.4 | 46.3 | 45.9 | 1208.9 | 7945.1 | 8642.2 |

Table 3: Training cost of Transformer (Trm), Transformer with FlashAttention (Trm+FlashAttn), and RetNet. We report memory consumption and training throughput (word per second; wps).

et al., 2012), and StoryCloze (SC; Mostafazadeh et al. 2017). The accuracy numbers are consistent with language modeling perplexity presented in Figure 3. RetNet achieves comparable performance with Transformer on zero-shot and in-context learning settings.

## 3.3 TRAINING COST

As shown in Table 3, we compare the training speed and memory consumption of Transformer and RetNet, where the training sequence length is 8192. We also compare with FlashAttention (Dao et al., 2022a), which improves speed and reduces GPU memory IO by recomputation and kernel fusion. In comparison, we implement RetNet using vanilla PyTorch code, and leave kernel fusion or FlashAttention-like acceleration for future work. We use chunkwise recurrent representation of retention as described in Equation (7). The chunk size is set to 512. We evaluate the results with eight Nvidia A100-80GB GPUs, because FlashAttention is highly optimized for A100. Tensor parallelism is enabled for 6.7B and 13B models.

Experimental results show that RetNet is more memory-efficient and has higher throughput than Transformers during training. Even compared with FlashAttention, RetNet is still competitive in terms of speed and memory cost. Moreover, without relying on specific kernels, it is easy to train RetNet on other platforms efficiently. For example, we train the RetNet models on an AMD MI200 cluster with decent throughput. It is notable that RetNet has the potential to further reduce cost via advanced implementation, such as kernel fusion.

## 3.4 INFERENCE COST

As shown in Figure 4, we compare memory cost, throughput, and latency of Transformer and RetNet during inference. Transformers reuse KV caches of previously decoded tokens. RetNet uses the recurrent representation as described in Equation (6). We evaluate the 6.7B model on the A100-80GB GPU. Figure 4 shows that RetNet outperforms Transformer in terms of inference cost.

**Memory** As shown in Figure 4a, the memory cost of Transformer increases linearly due to KV caches. In contrast, the memory consumption of RetNet remains consistent even for long sequences, requiring much less GPU memory to host RetNet. The additional memory consumption of RetNet is almost negligible (i.e., about 3%) while the model weights occupy 97%.

**Throughput** As presented in Figure 4b, the throughput of Transformer drops along with the decoding length increases. In comparison, RetNet has higher and length-invariant throughput during decoding, by utilizing the recurrent representation of retention.

**Latency** Latency is an important metric in deployment, which greatly affects user experience. We report decoding latency in Figure 4c. Experimental results show that increasing batch size renders Transformer's latency larger. Moreover, the latency of Transformers grows faster with longer input. In order to make latency acceptable, we have to restrict the batch size, which harms the overall inference throughput of Transformers. By contrast, RetNet's decoding latency outperforms Transformers and keeps almost the same across different batch sizes and input lengths.

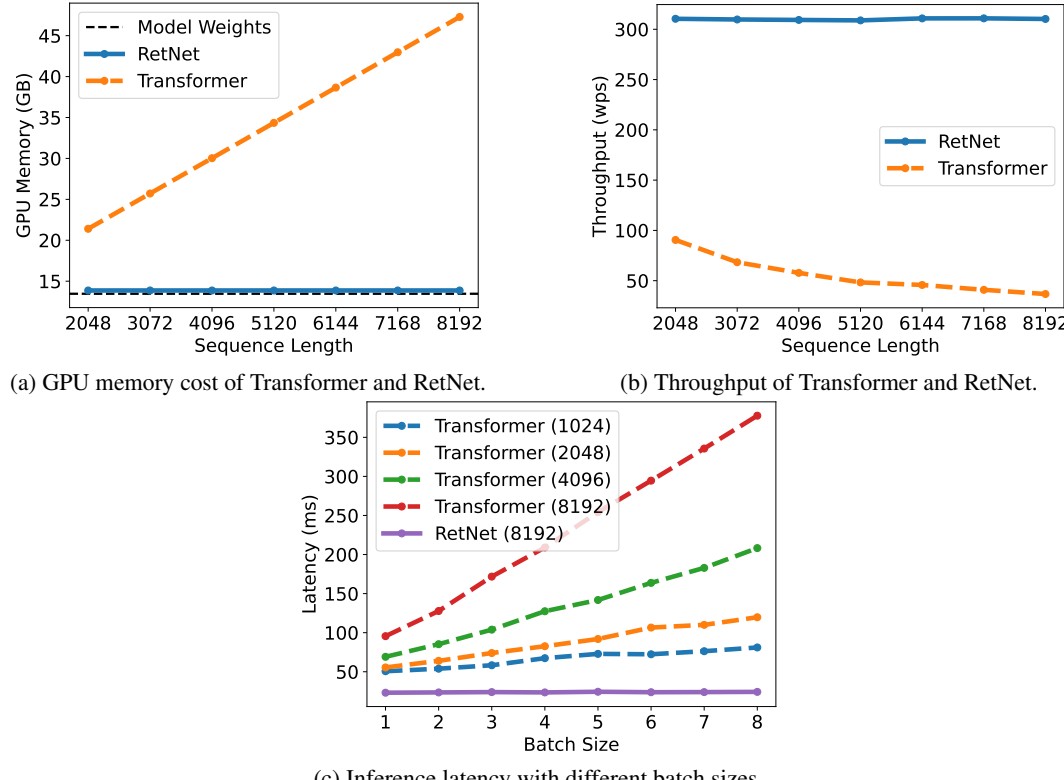

(a) GPU memory cost of Transformer and RetNet.

(b) Throughput of Transformer and RetNet.

(c) Inference latency with different batch sizes.

Figure 4: Inference cost of Transformer and RetNet with a model size of 6.7B. RetNet outperforms Transformers in terms of memory consumption, throughput, and latency.

| Method | In-Domain | PG22 | QMSum | GovReport | SummScreen |
|---|---|---|---|---|---|
| RWKV | 30.92 | 51.41 | 28.17 | 19.80 | 25.78 |
| H3 | 29.97 | 49.17 | 24.29 | 19.19 | 25.11 |
| Hyena | 32.08 | 52.75 | 28.18 | 20.55 | 26.51 |
| Linear Transformer | 40.24 | 63.86 | 28.45 | 25.33 | 32.02 |
| RetNet | **26.05** | **45.27** | **21.33** | **16.52** | **22.48** |

Table 4: Perplexity results on language modeling. RetNet outperforms other architectures on both the in-domain evaluation set and various out-of-domain corpora.

### 3.5 COMPARISON WITH TRANSFORMER VARIANTS

Apart from Transformer, we compare RetNet with various efficient Transformer variants, including Linear Transformer (Katharopoulos et al., 2020), RWKV (Peng et al., 2023), H3 (Dao et al., 2022b), and Hyena (Poli et al., 2023). All models have 200M parameters with 16 layers and a hidden dimension of 1024. For H3, we set the head dimension as 8. For RWKV, we use the TimeMix module to substitute self-attention layers while keeping FFN layers consistent with other models for fair comparisons. We train the models with 10k steps with a batch size of 0.5M tokens. Most hyperparameters and training corpora are kept the same as in Section 3.1.

Table 4 reports the perplexity numbers on the in-domain validation set and other out-of-domain corpora, e.g., Project Gutenberg 2019-2022 (PG22; Sun et al. 2023), QMSum (Zhong et al., 2021), GovReport (Huang et al., 2021), SummScreen (Chen et al., 2021; Shaham et al., 2022). Overall, RetNet outperforms previous methods across different datasets. RetNet not only achieves better evaluation results on the in-domain corpus but also obtains lower perplexity on several out-of-domain

| Method | In-Domain | PG22 | QMSum | GovReport | SummScreen |
|---|---|---|---|---|---|
| RetNet | **26.05** | **45.27** | **21.33** | **16.52** | **22.48** |
| $-$ swish gate | 27.84 | 49.44 | 22.52 | 17.45 | 23.72 |
| $-$ GroupNorm | 27.54 | 46.95 | 22.61 | 17.59 | 23.73 |
| $-$ $\gamma$ decay | 27.86 | 47.85 | 21.99 | 17.49 | 23.70 |
| $-$ multi-scale decay | 27.02 | 47.18 | 22.08 | 17.17 | 23.38 |
| Reduce head dimension | 27.68 | 47.72 | 23.09 | 17.46 | 23.41 |

Table 5: Ablation results on in-domain and out-of-domain corpora.

datasets. The favorable performance makes RetNet a strong successor to Transformer, besides the benefits of significant cost reduction (Sections 3.3 and 3.4).

In addition, we discuss the training and inference efficiency of the compared methods. Let $d$ denote the hidden dimension, and $n$ the sequence length. For training, RWKV's token-mixing complexity is $O(dn)$ while Hyena's is $O(dn \log n)$ with Fast Fourier Transform acceleration. The above two methods reduce training FLOPS via employing element-wise operators to trade-off modeling capacity. In comparison with retention, the chunk-wise recurrent representation is $O(dn(b + h))$, where $b$ is the chunk size, $h$ is the head dimension, and we usually set $b = 512, h = 256$. For either large model size (i.e., larger $d$) or sequence length, the additional $b + h$ has negligible effects. So the RetNet training is quite efficient without sacrificing the modeling performance. For inference, among the compared efficient architectures, Hyena has the same complexity (i.e., $O(n)$ per step) as Transformer while the others can perform $O(1)$ decoding.

### 3.6 ABLATION STUDIES

We ablate various design choices of RetNet and report the language modeling results in Table 5. The evaluation settings and metrics are the same as in Section 3.5.

**Architecture** We ablate the swish gate and GroupNorm as described in Equation (8). Table 5 shows that the above two components improve performance. First, the gating module is essential for enhancing non-linearity and improving model capability. Notice that we use the same parameter allocation as Transformers after removing the gate. Second, group normalization in retention balances the variances of multi-head outputs, which improves training stability and language modeling results.

**Multi-Scale Decay** Equation (8) shows that we use different $\gamma$ as the decay rates for the retention heads. In the ablation studies, we examine removing $\gamma$ decay (i.e., "$-\gamma$ decay") and applying the same decay rate across heads (i.e., "$-$ multi-scale decay"). Specifically, ablating $\gamma$ decay is equivalent to $\gamma = 1$. In the second setting, we set $\gamma = 127/128$ for all heads. Table 5 indicates that both the decay mechanism and using multiple decay rates can improve the language modeling performance.

**Head Dimension** As indicated by the recurrent perspective of Equation (1), the head dimension implies the memory capacity of hidden states. In ablation, we reduce the default head dimension from 256 to 64, i.e., 64 for queries and keys, and 128 for values. We keep the hidden dimension $d_{\text{model}}$ the same. Table 5 shows that the larger head dimension achieves better performance.

### 4 CONCLUSION

In this work, we propose retentive networks (RetNet) for sequence modeling, which enables various representations, i.e., parallel, recurrent, and chunkwise recurrent. RetNet achieves significantly better inference efficiency (in terms of memory, speed, and latency), favorable training parallelization, and competitive performance compared with Transformers. The above advantages make RetNet an ideal successor to Transformers for large language models, especially considering the deployment benefits brought by the $O(1)$ inference complexity. In the future, we would like to scale up RetNet in terms of model size and training steps. In addition, we are interested in deploying RetNet models on various edge devices, such as mobile phones.

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

# A  HYPERPARAMETERS

| Hyperparameters | 1.3B | 2.7B | 6.7B |
|---|---|---|---|
| Layers | 24 | 32 | 32 |
| Hidden size | 2048 | 2560 | 4096 |
| FFN size | 4096 | 5120 | 8192 |
| Heads | 8 | 10 | 16 |
| Learning rate | $6 \times 10^{-4}$ | $3 \times 10^{-4}$ | $3 \times 10^{-4}$ |
| LR scheduler | | Linear decay | |
| Warm-up steps | | 375 | |
| Tokens per batch | | 4M | |
| Adam $\beta$ | | (0.9, 0.98) | |
| Training steps | | 25,000 | |
| Gradient clipping | | 2.0 | |
| Dropout | | 0.1 | |
| Weight decay | | 0.01 | |

Table 6: Hyperparamters used for the models in Section 3.

# B  RESULTS WITH DIFFERENT CONTEXT LENGTHS

As shown in Table 7, we report language modeling results with different context lengths. In order to make the numbers comparable, we use 2048 text chunks as evaluation data and only compute perplexity for the last 128 tokens. Experimental results show that RetNet outperforms Transformer across different context lengths. Besides, RetNet can utilize longer context for better results.

| Model | 512 | 1024 | 2048 |
|---|---|---|---|
| Transformer | 13.55 | 12.56 | 12.35 |
| RetNet | 13.09 | 12.14 | 11.98 |

Table 7: Language modeling perplexity of RetNet and Transformer with different context length. The results show that RetNet has a consistent advantage across sequence length.

# C  INFERENCE COST OF GROUPED-QUERY RETENTION

We compare with grouped-query attention (Ainslie et al., 2023) and evaluate the method in the context of RetNet. Grouped-query attention makes a trade-off between performance and efficiency, which has been successfully verified in LLaMA2 34B/70B (Touvron et al., 2023b). The method reduces the overhead of key/value cache during inference. Moreover, the performance of grouped-query attention is better than multi-query attention (Shazeer, 2019), overcoming the quality degradation brought by using one-head key value.

As shown in Table 8, we compare the inference cost with grouped-query attention and apply the method for RetNet. For the LLaMA2 70B model, the number of key/value heads is reduced by $8\times$, where the query head number is 64 while the key/value head number is 8. For RetNet-70B, the parameter allocation is identical to LLaMA (Touvron et al., 2023a), where the dimension is 8192, and the head number is 32 for RetNet. For RetNet-70B-GQ2, the key-value head number is 16, where grouped-query retention is applied. We run the inference with four A100 GPUs without quantization.

When the batch size is 256, LLaMA2 runs out of memory while RetNet without group query still has a high throughput. When equipped with grouped-query retention, RetNet-70B achieves 38% acceleration and saves 30% memory.

We evaluate LLaMA2 under 2k and 8k lengths separately. The batch size is decreased to 8 so that LLaMA2 can be run without out of memory. Table 8 shows that the inference cost of Transformers increases with the sequence length. In contrast, RetNet is length-invariant. Moreover, RetNet-70B-GQ2 achieves better latency, throughput, and GPU memory than LLaMA2-70B-2k/8k equipped with grouped-query attention. Notice that evaluation metrics are averaged over positions of different sequence lengths for fair comparison, rather than only considering the inference cost of maximum length.

| Model | Batch Size | Latency (ms)↓ | Throughput (wps)↑ | Memory (GB)↓ |
|---|---|---|---|---|
| LLaMA2-70B-2k | 256 | — | — | OOM |
| LLaMA2-70B-8k | 256 | — | — | OOM |
| RetNet-70B | 256 | 639.1 | 410.19 | 72.469 |
| RetNet-70B-GQ2 | 256 | 461.8 | 567.66 | 52.726 |
| LLaMA2-70B-2k | 8 | 184.5 | 44.42 | 33.374 |
| LLaMA2-70B-8k | 8 | 277.7 | 29.50 | 37.386 |
| RetNet-70B-GQ2 | 8 | 106.2 | 77.02 | 32.301 |

Table 8: Inference cost of RetNet and LLaMA2-70B with difference batch size and length. LLaMA2-70B is equipped with grouped-query attention, reducing key/value heads by $8\times$. "-GQ2" means grouped-query retention, which reduces half of key/value heads. "-2k" and "-8k" indicate sequence length for LLaMA2, while RetNet is length-invariant. RetNet is capable of large-batch inference and is favourable in terms of latency, throughput, and GPU memory.

# D    PSEUDO CODE OF RETENTION

```
def ParallelRetention(
    q, # bsz * num_head * len * qk_dim
    k, # bsz * num_head * len * qk_dim
    v, # bsz * num_head * len * v_dim
    decay_mask # num_head * len * len
    ):
    retention = q @ k.transpose(-1, -2)
    retention = retention * decay_mask
    output = retention @ v
    output = group_norm(output)
    return output
```

```
def RecurrentRetention(
    q, k, v, # bsz * num_head * len * qkv_dim
    past_kv, # bsz * num_head * qk_dim * v_dim
    decay # num_head * 1 * 1
    ):
    current_kv = decay * past_kv + k.unsqueeze
        (-1) * v.unsqueeze(-2)
    output = torch.sum(q.unsqueeze(-1) *
        current_kv, dim=-2)
    output = group_norm(output)
    return output, current_kv
```

```
def ChunkwiseRetention(
    q, k, v, # bsz * num_head * chunk_size * qkv_dim
    past_kv, # bsz * num_head * qk_dim * v_dim
    decay_mask, # num_head * chunk_size * chunk_size
    chunk_decay, # num_head * 1 * 1
    inner_decay, # num_head * chunk_size
    ):
    retention = q @ k.transpose(-1, -2)
    retention = retention * decay_mask
    inner_retention = retention @ v
    cross_retention = (q @ past_kv) * inner_decay
    retention = inner_retention + cross_retention
    output = group_norm(retention)
    current_kv = chunk_decay * past_kv + k.transpose(-1, -2) @ v
    return output, current_kv
```

Figure 5: Pseudocode for the three computation paradigms of retention.

