# OpenReview forum: "Retentive Network: A Successor to Transformer for Large Language Models"
_ICLR.cc/2024/Conference — Submitted to ICLR 2024_

### Official Review · Reviewer_8FpU · 2023-11-01

**Soundness:** 2 fair
**Presentation:** 2 fair
**Contribution:** 2 fair
**Rating:** 3
**Confidence:** 5

**Summary:**

The authors propose a network called RetNet for language modeling, which has a linear training complexity and constant inference complexity.

**Strengths:**

This paper proposes a new architecture called RetNet, which has linear training complexity and constant inference speed.

**Weaknesses:**

The main weakness with this paper are overclaiming and lack of citations, which can be misleading for readers. For example, the claim in Figure 1 that "RetNet makes the 'impossible triangle' possible" is an absolute overclaim because the paper lacks validation with larger models and comparison with open-source Transformer models. On the other hand, the authors claim that RWKV and Linear Attention perform poorly, but according to [1], [2], their performance can be on par with Transformers. In Section 2, the authors introduce a new term called "Retention," but this is essentially the same as Linear Attention without the denominator, which has already been proposed in [2], [3]. Additionally, the use of EMA in MEGA and RWKV has been implemented, but the authors fail to cite these works.


[1] Bo Peng, Eric Alcaide, Quentin Anthony, Alon Albalak, Samuel Arcadinho, Huanqi Cao, Xin Cheng, Michael Chung, Matteo Grella, Kranthi Kiran G. V., Xuzheng He, Haowen Hou, Przemyslaw Kazienko, Jan Kocon, Jiaming Kong, Bartlomiej Koptyra, Hayden Lau, Krishna Sri Ipsit Mantri, Ferdinand Mom, Atsushi Saito, Xiangru Tang, Bolun Wang, Johan S. Wind, Stanislaw Wozniak, Ruichong Zhang, Zhenyuan Zhang, Qihang Zhao, Peng Zhou, Jian Zhu, and Rui-Jie Zhu. RWKV: reinventing rnns for the transformer era. CoRR, abs/2305.13048.

[2] Zhen Qin, Xiaodong Han, Weixuan Sun, Dongxu Li, Lingpeng Kong, Nick Barnes, and Yiran Zhong. The devil in linear transformer. In Proceedings of the 2022 Conference on Empirical Methods in Natural Language Processing, pages 7025–7041, Abu Dhabi, United Arab Emirates, Dec. 2022. Association for Computational Linguistics.

[3] * Huanru Henry Mao: “Fine-Tuning Pre-trained Transformers into Decaying Fast Weights”, Proceedings of the 2022 Conference on Empirical Methods in Natural Language Processing, pages 10236–10242, Abu Dhabi, United Arab Emirates, Dec. 2022. Association for Computational Linguistics.

**Questions:**

1. Figure 1 is an absolute overclaim because RWKV[1] and H3[2] have already demonstrated models at the billion-scale level that can achieve performance comparable to Transformers, with parallel training and constant inference. I suggest the authors remove this figure as it could mislead readers.
2. The description of RWKV in Table 1 is completely wrong. According to RWKV[1] and the description in [2], RWKV can indeed be computed in parallel. On the other hand, according to the RWKV paper, its performance is comparable to Transformers, so describing its performance as ✔ is also inaccurate. Overall, Table 1 is highly misleading and can affect the authors' judgment of model performance. I suggest the authors reorganize this table accordingly.
3. The form of Equation 1 is similar to RFA-GATE presented in [7], but the authors did not cite these articles throughout the paper.
4. The form of GroupNorm in Equation 8 is consistent with the NormAttention proposed in [5], but the author does not cite it at all.
5. There is a mistake in the description of the Linear Attention section in Section 2.4. Firstly, Linear Attention refers to the use of the Right-product trick to reduce complexity, but it does not necessarily imply approximation. For example, [4], [5], and [6] do not involve an approximation approach like softmax.
6. The statement "However, linear attention struggles to effectively encode position information, rendering the models less performant" should be supported by a reference since there could be various reasons for the poor performance. Moreover, the issue of the performance of Linear Attention has already been addressed in [5], where they propose a solution.
7. The discussion about MEGA and MEGA-chunk is missing, as they are similar to RetNet in utilizing the EMA technique.
8. Table 2 lacks comparison with open-source models. Firstly, the configuration of the Transformer is not mentioned, whether it is based on GPT2 architecture or Llama architecture. Additionally, there is no information provided regarding the parameter count or training data. On the other hand, there is no comparison with open-source models such as Bloom, Pythia, GPT-Neo, or RWKV. Comparing with these open-source models would allow readers to better understand the performance level of the proposed model.
9. The evaluation scope is too limited, for example, MMLU is not assessed.
10. It is indeed odd that the evaluation datasets in Tables 4 and 5 are inconsistent with Table 3. There should be further explanations provided to clarify this discrepancy.
11. There is a lack of ablation analysis on the architectural design, such as why the dimension of $W_v$ is chosen as d * 2d. On the other hand, an ablation for adding head dimension should be included in Table 5.

[1] Bo Peng, Eric Alcaide, Quentin Anthony, Alon Albalak, Samuel Arcadinho, Huanqi Cao, Xin Cheng, Michael Chung, Matteo Grella, Kranthi Kiran G. V., Xuzheng He, Haowen Hou, Przemyslaw Kazienko, Jan Kocon, Jiaming Kong, Bartlomiej Koptyra, Hayden Lau, Krishna Sri Ipsit Mantri, Ferdinand Mom, Atsushi Saito, Xiangru Tang, Bolun Wang, Johan S. Wind, Stanislaw Wozniak, Ruichong Zhang, Zhenyuan Zhang, Qihang Zhao, Peng Zhou, Jian Zhu, and Rui-Jie Zhu. RWKV: reinventing rnns for the transformer era. CoRR, abs/2305.13048.

[2] Tri Dao, Daniel Y. Fu, Khaled Kamal Saab, Armin W. Thomas, Atri Rudra, and Christopher Ré. Hungry hungry hippos: Towards language modeling with state space models. CoRR, abs/2212.14052, 2022.

[3] Eric Martin and Chris Cundy. 2017. Parallelizing linear recurrent neural nets over sequence length. ArXiv, abs/1709.04057.

[4] Zhen Qin, Weixuan Sun, Hui Deng, Dongxu Li, Yunshen Wei, Baohong Lv, Junjie Yan, Lingpeng Kong, and Yiran Zhong. cosformer: Rethinking softmax in attention. In ICLR, 2022.

[5] Zhen Qin, Xiaodong Han, Weixuan Sun, Dongxu Li, Lingpeng Kong, Nick Barnes, and Yiran Zhong. The devil in linear transformer. In Proceedings of the 2022 Conference on Empirical Methods in Natural Language Processing, pages 7025–7041, Abu Dhabi, United Arab Emirates, Dec. 2022. Association for Computational Linguistics.

[6] Angelos Katharopoulos, Apoorv Vyas, Nikolaos Pappas, and François Fleuret. Transformers are rnns: Fast autoregressive transformers with linear attention. In Proceedings of the 37th International Conference on Machine Learning, ICML 2020, 13-18 July 2020, Virtual Event, volume 119 of Proceedings of Machine Learning Research, pages 5156–5165. PMLR, 2020.

[7] Hao Peng, Nikolaos Pappas, Dani Yogatama, Roy Schwartz, Noah Smith, and Lingpeng Kong. Random feature attention. In International Conference on Learning Representations, 2020

---

> ### Author Response · Authors · 2023-11-22
>
> Q1: “Impossible Triangle” is an absolute overclaim because RWKV and H3 have already demonstrated models are comparable to Transformers
>
> A1: The claim is fair enough. The “comparable performance” means that the models achieve similar results under the same setting (e.g., #parameters, and training corpus). For example, previous comparisons use Transformers with absolute position while the compared methods benefit from relative position modeling. Moreover, in H3 paper, the comparable results are in hybrid settings (i.e., combine H3 and Transformer layers), but we don’t add any Transformer layers. We conducted various controlled experiments (with matched #parameters and using the same training corpus) to compare different architectures. We are confident that the claim holds well. The experiments in Table 4 also show that previous methods still have a big gap.
>
>
> Q2: RWKV can indeed be computed in parallel.
>
> A2: We give a clear definition on “training parallelization” in the caption of Table 1, which is discussed from the sequential perspective. “∗”: whether the training implementation is sequentially parallelized, although RWKV uses channel-wise parallelism. As stated in A1, RWKV’s performance is actually not comparable with Transformers according to our experiments (i.e., same #parameters, same data, and with relative position modelings). So, the statement of RWKV in Table 1 is fair enough.
>
> Q3: The form of Equation 1 is similar to RFA-GATE
>
> A3: Eq. (1) and RFA-GATE are not the same. Linear Attention is a candidate for RFA-Gate, where they both have denominators. In contrast, Eq. (1) does not include $A$. We can add the citation as suggested.
>
> Q4: The form of GroupNorm in Equation 8 is consistent with the NormAttention.
>
> A4: GroupNorm is significantly better than LayerNorm for multi-scale retention, as different heads have different statistic information. The implementation is different from NormAttention here. We can discuss the difference in the paper.
>
>
> Q5: There is a mistake in the description of the Linear Attention section in Section 2.4.
>
> A5: “approximating softmax” means that previous methods also encode a “probability distribution” on different positions, which is valid in [4] and [6]. Moreover, “Linear Attention” is not “Linearized Attention”, it stands for the paper “Transformers are RNNs: Fast autoregressive transformers with linear attention”.
>
> Q6: The statement "However, linear attention struggles to effectively encode position information, rendering the models less performant" should be supported by a reference since there could be various reasons for the poor performance.
>
> A6: Previous methods use kernel methods on query and key, which brings difficulty on relative position such as RoPE, where RoPE produces negative values for denominators. Otherwise the attention scores tend to become too flatten. [5] uses block softmax attention as an ad-hoc solution to make attention local.
>
> Q7: The discussion about MEGA and MEGA-chunk is missing, as they are similar to RetNet in utilizing the EMA technique.
>
> A7: RetNet has no direct connection with EMA technique. Instead, EMA is a weaker version of S4, so it’s convincing to compare with H3, which is a stronger baseline.

---

> ### Author Response · Authors · 2023-11-22
>
> Q8: Table 2 lacks comparison with open-source models.
>
> A8: We use Transformers with RoPE relative positions, which is a relatively strong baseline. Compared with LLaMA, there are only two differences: RMSNorm and SwiGLU. Specifically, RMSNorm typically improves stability instead of performance. Both RMSNorm and SwiGLU are orthogonal to our work. As long as the comparison settings are rigor and fair, the conclusions are solid. I totally understand you would like to compare different checkpoints, although they usually used different training data, data preprocessing pipelines, architecture configurations, number of training tokens, and hyperparameters. The paper focuses on fair comparisons rather than chasing for benchmarks.
>
> Q9: The evaluation scope is too limited.
>
> A9:
> The language modeling perplexity correlates very well with different tasks. Moreover, the language models have unified all tasks as generation. We report language modeling perplexity and various downstream task performance. We didn’t evaluate translation because our training corpus contains English data instead of multilingual data. We additionally evaluate one-shot performance on two open-ended question answering tasks with 7B models as follows. Notice that we report the recall metric in the table, i.e., whether the answers are contained in the generated response.
>
> Dataset / Transformer / RetNet
>
> Squad / 67.7 / 72.7
>
> WebQS / 36.4 / 40.4
>
> Q10: the evaluation datasets in Tables 4 and 5 are inconsistent with Table 3
>
> A10: We divide our evaluation into two groups: large scale comparison with Transformers and small size with other baselines. Evaluating perplexity is a stable and predictable metric for language modeling. We mainly report end-task accuracy numbers for larger-size models while reporting perplexity for small-size models. As indicated in previous work ([2304.15004] Are Emergent Abilities of Large Language Models a Mirage?), perplexity is smoother and more robust than accuracy for small models.
>
> Q11: There is a lack of ablation analysis on the architectural design.
>
> A11: The dimension modification of $W_v$ and FFNs aim to align the same parameter count with Transformers. Besides, increasing head dimension is harmful for inference efficiency. So the current setting is more scientific for evaluation.

---

### Official Review · Reviewer_Y3WY · 2023-11-02

**Soundness:** 2 fair
**Presentation:** 3 good
**Contribution:** 2 fair
**Rating:** 5
**Confidence:** 2

**Summary:**

This paper proposes a Retentive Network as a foundational architecture for large language models. Compared to Linear Transformers, Recurrent Networks, and Transformers, Retentive Networks can train in parallel, are low cost in inference, and have high quality. This paper discusses an interesting architectural design space where: 1) Transformer model families are inference inefficient while can be trained in parallel and are high quality; 2) Recurrent Networks on the other hand, can run inference relatively at lower cost but cannot be trained efficiently. The proposed Retentive Networks achieves all three by introducing a multi-scale retention mechanism to substitute multi-head attention. The method is greatly simplified without key-value cache tricks. The chunkwise recurrent representation can perform efficient long-sequence modeling. Empirical resuolts decodes 8x faster and saves 70% of memory compared to Transformer. During training, it is 7x faster with 20%-50% lower memory.

**Strengths:**

- The proposed retention mechanism has a dual form of recurrence and parallelism.

- The paper is well written and formalized properly.

**Weaknesses:**

- The paper is not clear to the reviewer why these two forms in Figure 2 (a) and Figure 2 (b) are equivalent.

- Are there any theoretical proof that retention is more capable than full attention?

- The paper is not clear why in Figure 3, RetNet is more effective in the large model regime. According to some prior work [1][2], two model architectures should not cross over when scaling the model up in log scale.

- Results are only provided on classification and summarization tasks, not generative tasks like translation and question answering. This shows limited generalization of the model architecture.

[1]: https://arxiv.org/abs/2001.08361
[2]: https://arxiv.org/abs/1712.00409

**Questions:**

1. Why would RetNet be better than full attention based transformer in terms of quality? Full attention should be more capable than recurrent networks.

2. Do you have results on NLG tasks? Refer to GPT3 paper and report some numbers on generative tasks.

3. The model scaling curve in Figure 3 looks odd to me. Why would the two lines cross over? It does not align with other empirical findings in early papers.

---

> ### Author Response · Authors · 2023-11-22
>
> Q1: The paper is not clear to the reviewer why these two forms in Figure 2 (a) and Figure 2 (b) are equivalent.
>
> A1: The equivalency between parallel and recurrent form is discussed mathematically from Eq. (1) to Eq. (6), where Figure 2a is for Eq. (5), and Figure 2b is for Eq. (6).
>
> Q2: Are there any theoretical proof that retention is more capable than full attention?
>
> A2: The comparable capabilities are evaluated empirically in the submission. For retention, we allow negative values for “retention scores”. In comparison, full attention only allows positive values for attention scores. If we consider finite arithmetic precision in practice, retention potentially utilizes the capacity more sufficiently without wasting the negative numerical ranges. The claim “Full attention should be more capable than recurrent networks” is not trivial because attention is often worse than RNN or S4 under many tasks, such as learning formal language.
>
> Q3: two model architectures should not cross over when scaling the model up in log scale.
>
> A3: Here we are comparing two different architectures in one figure. The curves do not cross over for the same architecture. But it doesn’t hold anymore if we compare different methods. If the learning curve is $f(m)=\alpha m^{\beta_g}+\gamma$, there are three variants $\alpha, \beta, \gamma$. When $f_1(m) = f_2(m) $ and $\beta_{g1} > \beta_{g2}$, it’s possible that two lines are cross over. Our scaling curves are not contradictory with early papers.
>
> Q4: Do you have results on generative tasks like translation and question answering?
>
> A4: The language modeling perplexity correlates very well with different tasks. Moreover, the language models have unified all tasks as generation. We report language modeling perplexity and various downstream task performance. We didn’t evaluate translation because our training corpus contains English data instead of multilingual data. We additionally evaluate one-shot performance on two open-ended question answering tasks with 7B models as follows. Notice that we report the recall metric in the table, i.e., whether the answers are contained in the generated response.
>
> Dataset / Transformer / RetNet
>
> Squad / 67.7 / 72.7
>
> WebQS / 36.4 / 40.4

---

> > ### Comment · Reviewer_Y3WY · 2023-12-01
> > **Thank you for the responses.**
> >
> > Aligned with other reviewers, I will keep my score unchanged. The rebuttal partially addressed my questions and concerns.

---

### Official Review · Reviewer_pKLR · 2023-11-02

**Soundness:** 3 good
**Presentation:** 2 fair
**Contribution:** 2 fair
**Rating:** 5
**Confidence:** 5

**Summary:**

This paper proposed a new network architecture, retentive network (RetNet) for large language modeling. Similarly as state space models (SSMs) and Linear Attention models, RetNet can be formulated both in a parallel and recurrent views, achieving parallel training and efficient inference/decoding.

The authors conducted experiments on large-scale pre-training form language modeling, with model size from 1.3B to 13B. The proposed RetNet obtained better zero/few-shot performance on benchmarks than standard Transformer. And the speed and memory efficiency of RetNet is also slightly better than Transformer with Flash Attention.

**Strengths:**

The RetNet is well-motivated and the equations are clear. The experiments are conducted with relatively large models.

**Weaknesses:**

There are several serious concerns about this paper:

1. Table 1 is mis-leading. If I understand correctly, the recurrent state $s_n$ in Eq (1) is in the shape of $d\times d$, where $d$ is the model dimension and is very large in practice (sometimes even larger than $N$). In S4 or other SSMs, the shape of the recurrent hidden state is $h\times d$ with relatively small $h$, e.g. $h=32$. However, in Table 1 the authors claimed the inference cost of RetNet is $O(1)$.

2. Table 3 is unclear. If I understand correctly, the parallel representation in Eq(5) is used for model training. However, the complexity of Eq(5) is almost the same with standard attention mechanism, with only difference that it does not need to compute softmax. Then why RetNet is faster and using less-memory than Flash Attention? Flash Attention also leveraged the block-wise computation of the attention matrix, which is similar to the chunkwise representation of RetNet. In addition, which version of Flash Attention was used in these experiments? Flash Attention v2 has been significantly optimized.

3. The experimental setting is unconvincing. Though the authors scaled-up RetNet to 13B, the total number of training tokens are only 100B. It is well-known that Transformer is data-eager, and it is unfair to compare with Transformer with relatively small training data. Moreover, the improved version of Transformer in Llama (with RMSNorm and SwiGLU) has achieved significantly better results than the standard Transformer. But all the comparisons in this paper are with standard Transformer.

4. In Table 1, the authors claimed that only Transformer and RetNet are achieving "good performance". However, there are no direct comparison of RetNet with other models in large-scale setting. The results in Table 4 were conducted with a small 200M model size and only 5B tokens.

**Questions:**

The parallel allocation adjustment, i.e setting $V$ twice dimension of $d$, has already been investigated in prior works, such as Flash[1] and Mega[2].


[1] Hua et al., Transformer Quality in Linear Time.

[2] Ma et al., Mega: Moving Average Equipped Gated Attention.

---

> ### Author Response · Authors · 2023-11-22
>
> Q1: The inference cost is not constant
>
> A1: Eq. (1) is discussed under one-head settings. We still use multi-head in Eq. (8), and the shape of $s_n$ is $h\times d$, where $h$ is query and key's head dimension. So the inference cost will be $O(1)$. Our inference cost experiment in Figure 4 demonstrates that.
>
> Q2: why RetNet is faster and using less-memory than Flash Attention?
>
> A2: For the comparison with FlashAttention, we use an optimized FlashAttention, which is better than FlashAttention1 and is comparable with FlashAttention2 for long sequences. Since the publication time of FlashAttention2 is later than ICLR submission deadline, we didn't compare with the official one in the submission.
>
> Q3: the total number of training tokens are only 100B. It’s not enough since Transformer is data-eager.
>
> A3: According to the Chinchilla scaling law, 7B model’s converge speed slows down at $FLOPs=10^{21}$. The flops for 100B tokens are more than $(7\times 10^ 9) \times (100 \times 10^ 9) \times 6 > 4\times 10^{21}$. So 100B tokens are enough for evaluating model performance.
>
> Q4: the improved version of Transformer in Llama (with RMSNorm and SwiGLU) has achieved significantly better results than the standard Transformer.
>
> A4: The LLaMA improvement is on training stability and FFN capacity, which is useful regardless of attention or other token-mixing methods. RMSNorm and SwiGLU can also be applied to RetNet. The comparisons under standard Transformer are fair.
>
> Q5: there are no direct comparison of RetNet with other models in large-scale setting.
>
> A5: The conclusions of comparing with other architectures are consistent even with larger-scale training. Scaling up every method went beyond our computation budget. As the de facto architecture is Transformer now, we spent more resources on the comparisons with Transformers, where we ran the experiments in large-scale settings.
>
> Q6: Missing citations.
>
> A6: We can add the suggested paper “[1] Hua et al., Transformer Quality in Linear Time” in the revision.

---

### Official Review · Reviewer_4ryG · 2023-11-03

**Soundness:** 3 good
**Presentation:** 2 fair
**Contribution:** 3 good
**Rating:** 6
**Confidence:** 5

**Summary:**

This work proposes RetNet, a model that promises training parallelism, low cost-inference and good performance. It is presented as a successor to Transformers and comes with very impressive results to back up this stance.

**Strengths:**

The experimental results of this approach are compelling. It appears that as model parameters increase beyod 2B, retentitive networks outperform Transformers on language modelling according to perplexity.

Inference no longer requires addititional KV cache allowing for O(1) inference cost.

RetNet allows for O(N) long-sequence memory complexity by accumulating into a buffer.

**Weaknesses:**

Novelty: this is essentially a transformer without the softmax and an added time decay. It just so happens that with scale, it appears that this difference does not hinder RetNet performance.

Clarity: The paper gets pretty dense at times affecting readability. Also figure 2b is hard to understand without the code.


This paper lacks a broader impacts section, its addition would strengthen the paper.

The code for the work appears to be closed source, given the overwhelmingly positive results.. it makes comparison and benchmarking against this approach difficult for other researchers in the field.

**Questions:**

So the results look overwhelmingly positive, could the authors discuss the shortcomings of this approach? On which tasks does this model not work well?


Will the code for this paper be open-sourced?

---

> ### Author Response · Authors · 2023-11-11
>
> 1. "this is essentially a transformer without the softmax and an added time decay." However, doing that in a naive way can't get a well-performed model. Previous works also made a similar contribution, but their performance still has a big gap in Table 4. We utilize new designs to achieve comparable performance, including gated multi-scale retention and group normalization.
> 2. Figure 2a and 2b correspond to Equation 5 and 6. Mathematical explanation may help.
> 3. RetNet's first shortcoming is that it is not strictly comparable under small. size, e.g. 1.3B. Besides, we work on GPT settings so it can't migrate to bidirectional transformers. Regular tasks in NLP almost follow the pre-training performance.
> 4. The code will be open-source definitely after review.

---

### Meta-Review · Area_Chair_cPaQ · 2023-12-04

**Metareview:**

This paper proposes RetNet, which is uses a  linear attention mechanism (with decay) instead of the usual softmax attention. The linearity makes it possible to formulate the model as an RNN, which can make inference more efficient. For training, the authors propose a chunking mechanism which makes the training subquadratic (as in an RNN) but still allows for some parallelization.

The main strength of this paper is simplicity of the approach combined with the empirical performance, which is shown at nontrivial scale (6B): RetNet shows that a simplified attention mechanism that requires O(1) memory during inference can be competitive with a Transformer, raising the question of whether the more expensive softmax attention is truly needed. However, there were several weaknesses that were highlighted, in particular that the (i) Transformer baseline may not be being strong enough, and that (ii) the claims being made in the paper (i.e., that RetNet is a successor to Transformers) are not fully supported by the current set of experiments (i.e., there is potential overclaiming).

I like the paper quite a bit and I think it has the potential to be an influential work. However, given the concerns raised by many of the reviewers, I cannot recommend this paper for acceptance in its current form.

**Justification For Why Not Higher Score:**

Baselines are potentially weak, and the paper's claimed contributions are not fully supported by the experimental results.

**Justification For Why Not Lower Score:**

N/A

---

### Decision · Program_Chairs · 2024-01-16

Reject